# Knowledge, Attitude, Risk Perception, and Health-Related Adaptive Behavior of Primary School Children towards Climate Change: A Cross-Sectional Study in China

**DOI:** 10.3390/ijerph192315648

**Published:** 2022-11-25

**Authors:** Yu Wang, Xinhang Zhang, Yonghong Li, Yanxiang Liu, Bo Sun, Yan Wang, Zhirong Zhou, Lei Zheng, Linxin Zhang, Xiaoyuan Yao, Yibin Cheng

**Affiliations:** 1China CDC Key Laboratory of Environment and Population Health, National Institute of Environmental Health, Chinese Center for Disease Control and Prevention, Beijing 100021, China; 2Tongzhou Center for Disease Control and Prevention, Beijing 101199, China; 3Public Meteorological Service Center of China Meteorological Administration, Beijing 100081, China

**Keywords:** behavior, adaptation, children, climate change, structural equation model (SEM)

## Abstract

Background: Children are disproportionately affected by climate change while evidence regarding their adaptive behavior and relevant influencing factors is limited. Objectives: We attempted to investigate health-related adaptive behavior towards climate change for primary school children in China and explore potential influencing factors. Methods: We conducted a survey of 8322 primary school children in 12 cities across China. Knowledge, attitude, risk perception, and adaptive behavior scores for children were collected using a designed questionnaire. Weather exposures of cities were collected from 2014 to 2018. We applied a multiple linear regression and mixed-effect regression to assess the influencing factors of adaptive behavior. We also used the structural equation model (SEM) to validate the theoretical framework of adaptive behavior. Results: Most children (76.1%) were aware of climate change. They mainly get information from television, smartphones, and the Internet. A 1 score increase in knowledge, attitude, and risk perception was associated with 0.210, 0.386, and 0.160 increase in adaptive behavior scores, respectively. Females and children having air conditioners or heating systems at home were positively associated with adaptive behavior. Exposure to cold and rainstorms increased the adaptive behavior scores, while heat exposure had the opposite effects. The SEM showed that knowledge was positively associated with attitude and risk perception. Knowledge, attitude, and risk perception corresponded to 31.6%, 22.8%, and 26.1% changes of adaptive behavior, respectively. Conclusion: Most primary school children in China were aware of climate change. Knowledge, attitude, risk perception, cold, and rainstorm exposure were positively associated with health-related adaptive behavior towards climate change.

## 1. Introduction

In the past ten years, 83% of natural disasters were attributed to weather- and climate-related events [1]. As our climate continues to change, the increasing frequency and intensity of extreme weather events have placed great threats to human health. Children are disproportionately affected by climate change given their unique physical, physiological, and behavioral patterns. Children undergo a critical and rapid development period but with an immature biological capacity to adapt, leaving windows of vulnerability to climate change stresses [2]. Children are also sensitive to climate change because of their high surface area-to-mass ratio, high exposures per unit body weight, and long lifespan exposures [3]. Many previous studies have documented elevated morbidity risks for children under extreme weather events [4,5,6,7]. To protect children from the detrimental effects of climate change, they need to be aware of the warming climate and take adaptive measures [8]. However, evidence is limited regarding their awareness and health-related adaptive behavior towards climate change.

An adaptive behavior towards climate change is any action that moderates the risk of climate change impacts. Health-related adaptive behaviors focus on behavior related to health and disease prevention such as wearing a hat when going outside on hot days. Current evidence has shown that adaptive behaviors towards climate change are influenced or motivated by several factors such as gender, income [9], knowledge [10,11], attitude [12], and risk perception [13], albeit with different definitions. Research into adaptive behaviors reported that risk perception was a key driver of taking adaptive behavior [11,14,15]. Experience of flood, heatwave, and other extreme weather events was also found to be positively associated with adaptive behavior [16,17]. However, the current evidence was mainly collected from residents in vulnerable areas [13,14,15] or from vulnerable populations like farmers [18] and workers [19]. Few studies have assessed the climate adaptation of children, particularly in developing countries. Therefore, there is great need to investigate the health-related adaptive behavior towards climate change for children and to identify potential influencing factors.

Behavioral change is not without hardship. Knowledge, attitude, and perceived risk sometimes failed to transfer to concrete behavioral change [18,20] as they were also modified by other factors or have interconnections. Many researchers have proposed theoretical frameworks to facilitate changing health-related behavior. The most commonly used theory is the knowledge, attitude, and practice (KAP) model [21,22] and the health belief model (HBM) [13,23]. KAP assumes that knowledge can lead to a change in attitude and then cause health-related behavior changes. HBM uses several components like perceived suspect, severity, benefit, and barriers to predict health-related behavior [24]. Due to the difference and coexistence of attitude and risk perception, a combination of both KAP and HBM may comprehensively reveal the pathways of behavior. We assumed that knowledge was positively associated with attitude and perception, all of which directly influenced the health-related adaptive behavior of children.

To this end, we tried to investigate knowledge, attitude, risk perception, and health-related adaptive behavior of primary school children towards climate change; explore potential influencing factors; and validate the theoretical framework of adaptive behavior concerning climate change.

## 2. Materials and Methods

### 2.1. Study Design

We carried out a cross-sectional study from November 2019 to January 2020 in 12 cities in China (Figure 1). We selected cities from a national program that investigated regional climate-sensitive diseases [25]. Selected cities were located in all 11 geo-meteorological regions and covered four of the five climate zones except tropical monsoon climate. We randomly selected two primary schools in each city and surveyed students in grades 3 to 5. The study was approved by the Ethics Committee of the National Institute of Environmental Health and the Chinese Center for Disease Control and Prevention (No. 201606).

### 2.2. Questionnaire Design and Collection

We developed the questionnaire mainly based on a KAP program in Jiangsu, China [26], a HBM program in Adelaide, Australia [13], and other literature on climate change [21,23,27]. The questions were revised by two experts in the field of non-communication diseases and one expert in each field of climate change, infectious diseases, and injury. We then optimized the statements and validated content reliability via a pilot study of around 20 students and 30 staff members from national and local Centers for Disease Control and Prevention. Finally, we included five parts in the questionnaire: general information, knowledge, attitude, risk perception, and health-related adaptive behavior towards climate change. Total scores of knowledge, attitude, risk perception, and adaptive behavior were 21, 20, 30, and 40, respectively. The questions and scores of the questionnaire are shown in Appendix A.

General information included gender, age, grade, air conditioners, and heating systems at home. The knowledge part of climate change refers to the impact and mitigation of climate change and recognition of measures in extreme weather and drowning prevention [26]. Seven statements were assigned in the knowledge part, in which statement 4 was negatively worded. For positively worded statements, responses of “True” or the selection of multiple-choice questions in impact and mitigation were given a score of “1” while responses of “False” or “None of them” were given a score of “0”. For the negatively worded statement, scores were assigned in reverse. If children did not know or had not heard about climate change, statements 1 and 2 were given a score of “0”.

The risk perception part is defined as the perceived risk and health threat of climate change, perceived barriers of mitigation, and perceived benefits from behavior changes [13,23]. The attitude part included the willingness to learn about climate change, health concerns in extreme weather, and willingness to change behavior to prevent diseases [21,26]. The adaptive behavior part focused on the health-related adaptive behavior towards climate change [26,27] including preventive measures in extreme weather and in relation to climate-related infectious diseases. We designed 6, 4, and 8 questions in the risk perception, attitude, and adaptive behavior part, respectively. All questions above were designed using a 5-point scale. Question 5 in the risk perception part and question 6 in the adaptive behavior part were negatively-worded. For the positively-worded questions, responses of “①” to “⑤” were given scores of “1” to “5” accordingly. For the negatively worded questions, scores were assigned in reverse.

Questionnaires were collected by trained staff at the local Centers for Disease Control and Prevention. Students were interviewed in class and answered the questions by themselves. Written consent was obtained from students and their parents. The integrity of questionnaires was examined before collection to control data quality. Data from the questionnaires was input twice by two staff separately using Epidata software to control input errors.

### 2.3. Exposure Data

To assess the impact of weather exposure and socioeconomic factors on the health-related adaptive behavior towards climate change, we acquired weather data during 2014–2018 from the nearest National Meteorological Station. Exposure to heat, cold, and rainstorms was defined as days with maximum temperature ≥ 35 °C, mean temperature ≤ 0 °C, and precipitation amount ≥ 50 mm, respectively. Those thresholds were used in the daily weather forecast. In addition, we collected the socioeconomic data, per capita disposable income (PCDI), of each city from the published local Statistical Communique on the 2019 Economic and Social Development.

### 2.4. Statistical Analysis

We used a simple and multiple linear regression to assess the effects of potential influencing factors. Due to the nature of city-level variables, we dichotomized heat, cold, rainstorm days, and PCDI into low (<median, a score of 0) and high (≥median, score of 1) levels. Gender (0 for male, 1 for female) and air conditioners and heating systems (0 for “do not have,” 1 for “have”) were all modeled as binary variables. The normality of the residents was respected due to a large number of observations. To avoid potential heteroscedasticity and autocorrelation, we estimated a robust standard error using heteroscedasticity and autocorrelation consistent (HAC) estimation [28]. To avoid dependence on high-level status, we also conducted a multiple linear mixed-effect regression with random effects for students and cities. The linear correlation and variance inflation factor (VIF) of the variables in the multiple regression were not large enough to render multicollinearity (Appendix A).

Due to the internal and complex relationship among potential influencing factors, we applied a structural equation model (SEM) to examine the effects of modifiable factors on the health-related adaptive behavior towards climate change. An SEM is a covariance-based multivariate technique that consists of measurement model and structural model. The SEM applies factor analysis to link latent variables with observed variables in the measurement model and conducts path analysis to examine the association between latent variables in the structural model. Based on the KAP and HBM model, we assumed that knowledge was positively associated with attitude and perception, all of which directly influenced health-related adaptive behavior. To test our assumption, we built a theoretical framework of adaptive behavior and validated it using the SEM. Root mean square error of approximation (RMSEA), comparative fit index (CFI), and Tucker-Lewis index (TLI) were used to assess the goodness-of-fit of the SEM. Cronbach’s alpha was used to assess the composite reliability. 

All data analyses were conducted using R software (version 4.0.2). Packages “sandwich,” “lme4,” and “lavaan” [29] were used to calculate the robust standard error, fit the mixed-effect model, and calculate the SEM, respectively.

## 3. Results

### 3.1. Descriptive Statistics

A total of 8480 questionnaires were collected, and 8322 (98.1%) were analyzed after removing incomplete and incorrect information. The number of questionnaires per city ranged from 635 to 903. Males and females accounted for 51.6% and 48.4% of the questionnaires, respectively. The mean age of children was 9.8 years. Nearly one-third of the children reported they have air conditioners (66.5%) and heating systems (69.7%) at home, respectively (Table 1).

Most of the children (76.1%) reported that they were aware of climate change. Among which, 57.0% get information from television, 42.7% from smartphones, and 32.2% from the Internet (Figure 2). The median score of knowledge, attitude, risk perception, and health-related adaptive behavior towards climate change was 10 [P25 (the 25th percentile), P75 (the 75th percentile): 5, 14], 18 (16, 19), 22 (20, 25), and 29 (25, 32), respectively (Table 1). At the city level, heat, cold, and rainstorm exposure during 2014–2018 were various. Knowledge and adaptive behavior scores varied across cities; however, risk perception and attitude scores remained almost the same (Table 2).

### 3.2. Influencing Factors of Health-Related Adaptive Behavior

We used a multiple linear regression to identify the influencing factors of health-related adaptive behavior (Table 3). After adjusting potential heteroscedasticity, a 1 score increase in knowledge, attitude, and risk perception was associated with 0.210, 0.386, and 0.160 increase in adaptive behavior scores, respectively. The standardized estimates were 0.200, 0.198, and 0.120. Females and children having air conditioners or heating systems at home were positively associated with adaptive behavior. Exposure to cold and rainstorms increased the adaptive behavior scores of children while heat exposure had negative effects. The impact of PCDI was not statistically significant in multiple regressions. A mixed-effect analysis with the student as a random effect showed constant results. A mixed-effect model with both student and city as random variables reduced the effects of having air conditioners or heating systems and made city-level weather exposures nonsignificant.

### 3.3. Verification of SEM of Health-Related Adaptive Behavior

We first modeled knowledge, attitude, risk perception, and health-related adaptive behavior as latent variables in the SEM. Questions for each part were observed variables. The original SEM with all observed variables had a RSMEA of 0.059, CFI of 0.783, a TLI of 0.759, and two nonsignificant unstandardized estimates (Figure 3A and Appendix A), which indicated that the model needs to be improved. To maintain the robustness of the SEM, we retained unstandardized estimated coefficients of 0.5 and above in the measurement model [15]. The final structure model is shown in Figure 3B with a RSMEA of 0.059, a CFI of 0.914, a TLI of 0.894, and all significant unstandardized estimates (*p* < 0.001) (Appendix A), which indicated the goodness of fit of the SEM [17]. Cronbach’s alphas were 0.846, 0.802, 0.664, and 0.668 for latent variables above, respectively.

Based on standardized estimates, knowledge, attitude, and risk perception were all positively-associated with adaptive behavior. In the original SEM, knowledge, attitude, and risk perception corresponded with 33.2%, 24.0%, and 24.9% changes in adaptive behavior. In the final SEM, they were responsible for 31.6%, 22.8%, and 26.1% changes of adaptive behavior, respectively. Knowledge was also positively associated with risk perception and attitude (Figure 3).

## 4. Discussion

We investigated knowledge, attitude, risk perception, and health-related adaptive behavior towards climate change for primary school children across China. Our study was conducted on a significant sample and evenly distributed across cities to cover all 11 geo-meteorological regions, which can reflect the awareness of climate change in children more exactly than current studies in one location or with a limited sample. We found that 76.1% of the children were aware of climate change. The awareness of climate change was unevenly distributed around the world with a range from 15% to 99% using data from the Gallup World Poll [30]. China indicated a 61.9% awareness of climate change from 7452 participants aged 15 and above [9]. Another national survey in China found that 93.4% of 4169 respondents have heard about climate change. The heterogeneity of awareness may be due to population distribution, survey period, and the description used in the questionnaire. Education, geographic location, and household income were also shown to affect climate change awareness in China [9]. Given the number of children and cities in the study, our results yield more reliable estimates.

We found that children get information about climate change mainly from television, smartphones, and the Internet. However, studies in Australia [31], the United States [20], and Ethiopia [32] showed that television, radio, and newspapers were the most common sources of climate information. The differences in sources of information implied that tailored education methods should be applied in different areas. Because of the widespread use of the Internet, 5G, and smartphones in China, e-learning programs may be more convenient and attractive for Chinese students, especially under the COVID-19 pandemic [33]. Moreover, a shorter and less time-consuming message delivery would be favored by children under the fierce competition of their daily homework and out-of-school learning.

We found that one standard deviation increase in knowledge, attitude, and risk perception was associated with 0.200, 0.198, and 0.120 standard deviation increase in health-related adaptive behavior towards climate change, respectively. Positive associations were validated by many previous studies across countries [13,15,21]. After controlling associations among knowledge, attitude, and risk perception in the SEM, they were responsible for 31.6%, 22.8%, and 26.1% changes in health-related adaptive behavior, respectively. The contribution of knowledge and risk perception to behavior change was higher in the SEM than in the MLR. A higher standardized estimate of knowledge implied that improving the knowledge of children may be a better choice to enhance their adaptive behavior. However, standardized estimates were not large enough to render substantial behavioral changes. A study in the United States indicated that only half of all respondents changed their behavior despite a diversity of adaptive information available [20]. A study in India showed that adaptive measures of farmers were mainly triggered by socioeconomic factors but not perceived climatic change [18]. One possible explanation is that climate change is relegated in public minds behind issues such as the economy for farmers and education for children [34]. Another explanation is that different types of perceived risk, like national/personal risk and long/short-term risk, lead to different types or levels of adaptive behavior. To tackle such a challenge, effective risk communications that speak to the interests and values of children or that facilitate through a peer-to-peer approach should be implemented [34]. In addition, comprehensive intervention including early warning messages coupled with adaptive behaviors should be applied and evaluated for their efficacy.

Though in part contradictory [35], the evidence available shows that people who have experienced extreme weather events are more likely to be concerned about climate change and undertake adaptive behaviors [16,36,37]. We found that cold and rainstorm experiences were positively associated with high levels of adaptive behavior, while heat exposure had negative effects. One possible interpretation is that heat exposure over time develops a sense of normalcy in the situation, especially in southern China. Children may not take additional adaptive behaviors to cope with climate change [35]. In addition, constant heat exposure may result in a seeming resilience to heat and issue fatigue [38]. In this scenario, climate knowledge and early warning messages should be routinely transmitted to children in high-risk areas before extreme weather events.

As children are disproportionately affected by climate change, equity issues need to be emphasized and included in climate change education and action. We found that children in low PCDI areas like Xining, Lasa, and Xiangtan have low levels of knowledge and adaptive behavior while they were not exposed to less extreme weather events. Though the PCDI levels showed a nonsignificant impact on adaptive behavior in multiple regressions, children may have to bear the brunt of climate change if they have less access to preventive facilities and measures. Moreover, some of the adaptive behavior and mitigation actions were not under the control of the children but rather their parents and schools, such as gathering information on climate change. The government and schools need to provide educational programs for children and child-to-parent intergenerational learning [39] to promote children’s right to participate and access information on climate change [40].

There are some limitations to our study. First, we focused on the health-related adaptive behavior towards climate change and did not include other relevant adaptive behavior like energy-saving measures. Second, certain adaptive behaviors of children may be outside their direct control, which may lead to an overestimated contribution of knowledge and other influencing factors. In recent decades, however, students may be more self-determined as they have more access to the Internet and knowledge. Third, although the SEM validated the complex relationship of knowledge, attitude, perception, and behavior, results could only be interpreted as an association instead of a causation. Further studies are warranted to investigate different types of behaviors and assess the effectiveness of education on the adaptive behavior concerning climate change.

## 5. Conclusions

With regard to the significant sample and evenly distributed cities of our study, we provided reliable evidence that most primary school children in China were aware of the warming climate. We found that knowledge, attitude, risk perception, cold, and rainstorm exposure were positively associated with their health-related adaptive behavior towards climate change. Further studies are warranted to assess the effects of educational programs on the awareness and adaptive behavior towards climate change for children.

## Figures and Tables

**Figure 1 ijerph-19-15648-f001:**
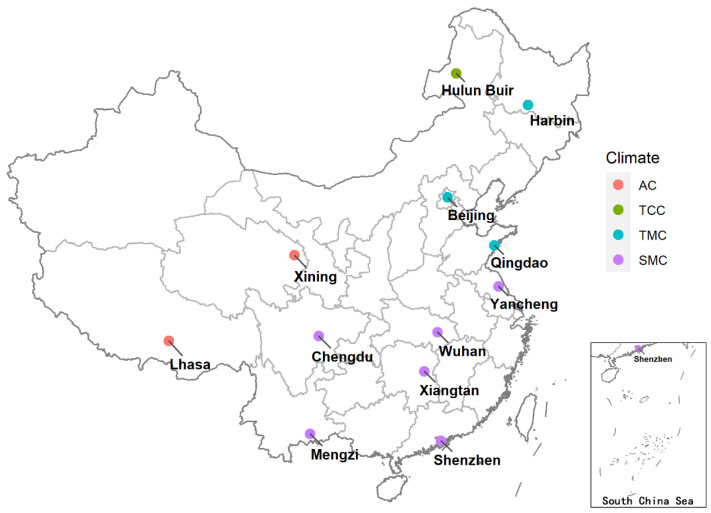
Location and climate of the study cities. AC, alpine climate. TCC, temperate continental climate. TMC, temperate monsoon climate. SMC, subtropical monsoon climate.

**Figure 2 ijerph-19-15648-f002:**
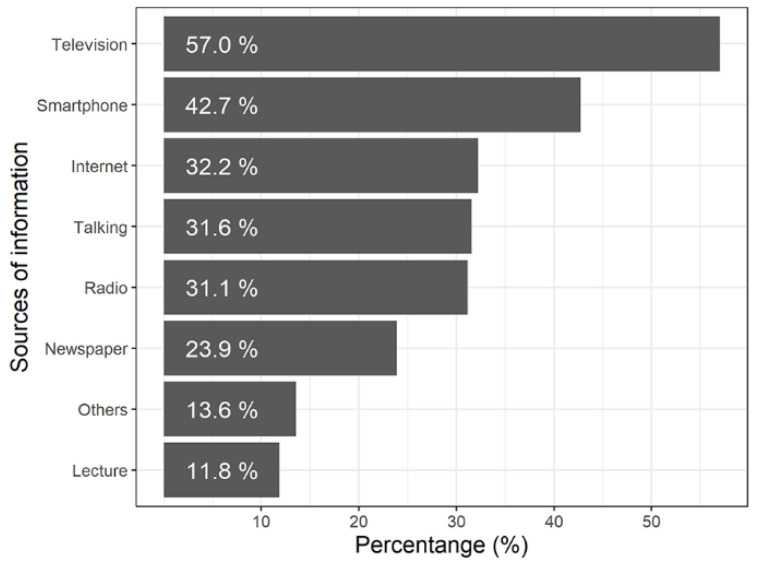
Sources of information about climate change.

**Figure 3 ijerph-19-15648-f003:**
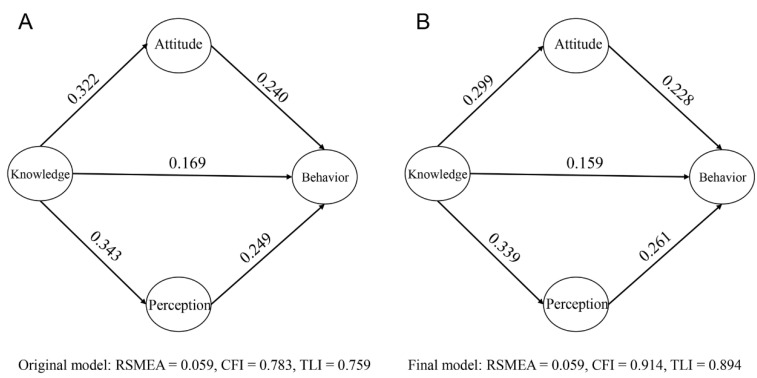
Validation of theoretical framework of adaptive behavior. Values refer to the standardized estimates. (**A**) refers to the original SEM. (**B**) refers to the final SEM after removing the unstandardized estimated coefficient of 0.5 and below.

**Table 1 ijerph-19-15648-t001:** Statistics summary.

	n (%)
Total	8322 (100.0)
Gender	
Male	4293 (51.6)
Female	4029 (48.4)
Age (mean ± SD)	9.8 ± 1.0
Grades	
Grade3	2939 (35.3)
Grade4	2935 (35.3)
Grade5	2448 (29.4)
Have air conditioners	5531 (66.5)
Have heating systems	5797 (69.7)
Aware of climate change	6333 (76.1)
Scores [median (P25, P75)]	
Knowledge	10 (5, 14)
Attitude	18 (16, 19)
Risk perception	20 (20, 25)
Adaptive behavior	29 (25, 32)

Note: SD, standard deviation. P25, the 25th percentile. P75, the 75th percentile.

**Table 2 ijerph-19-15648-t002:** Weather conditions and scores in each city.

City	Number of Questionnaires	Heat Days	Cold Days	Rainstorm Days	PCDI	Knowledge Score	Attitude Score	Risk Perception Score	Adaptive Behavior Score
Hulun Buir	686	13	826	86	30,570	12 (8, 17)	18 (17, 19)	23 (22, 26)	32 (28, 34)
Harbin	670	4	669	112	34,601	10 (6, 13)	18 (16, 19)	23 (20, 26)	29 (25, 32)
Beijing	714	56	438	8	40,067	10 (5, 13)	18 (16, 20)	22 (20, 24)	30 (27, 34)
Xining	719	0	556	55	28,189	7 (4, 10)	17 (15, 19)	21 (18, 23)	28 (25, 32)
Qingdao	645	3	129	126	45,452	10 (6, 13)	18 (16, 20)	23 (20, 26)	30 (27, 34)
Yancheng	635	58	62	95	32,096	8 (5, 12)	18 (16, 20)	22 (20, 25)	29 (25, 32)
Chengdu	695	26	6	179	40,371	12 (9, 15)	18 (16, 20)	21 (19, 24)	28 (25, 32)
Wuhan	903	108	30	21	46,010	12 (9, 16)	18 (16, 19)	22 (20, 24)	29 (26, 32)
Lasa	650	62	225	62	25,844	6 (4, 10)	16 (15, 19)	22 (20, 26)	27 (23, 29)
Xiangtan	670	176	40	82	26,647	7 (4, 12)	16 (14, 19)	21 (18, 24)	25 (22, 29)
Mengzi	660	0	0	7	32,398	10 (7, 15)	18 (16, 20)	22 (20, 25)	28 (25, 32)
Shenzhen	675	23	2	174	62,522	12 (9, 15)	18 (16, 20)	22 (20, 24)	30 (26, 33)

Note: Weather data was collected from 2014 to 2018. Per capita disposable income (PCDI) was collected in 2019. Scores refer to the median (P25, P75). P25, the 25th percentile. P75, the 75th percentile.

**Table 3 ijerph-19-15648-t003:** Influencing factors of adaptive behavior of climate change.

Variables	SLR		MLR		MLR ^a^		LMM ^b^		LMM ^c^	
	Estimates	*p* Value	Estimates	*p* Value	Estimates	*p* Value	Estimates	*p* Value	Estimates	*p* Value
Gender	1.071	<0.001	1.020	<0.001	1.020	<0.001	1.022	<0.001	1.041	<0.001
Grade	0.077	0.282	−0.167	0.010	−0.167	0.067	−0.167	0.010	−0.234	0.008
Knowledge	0.308	<0.001	0.210	<0.001	0.210	<0.001	0.210	<0.001	0.203	<0.001
Attitude	0.627	<0.001	0.386	<0.001	0.386	<0.001	0.386	<0.001	0.344	<0.001
Risk perception	0.361	<0.001	0.160	<0.001	0.160	<0.001	0.160	<0.001	0.168	<0.001
Heat exposure	−1.275	<0.001	−0.814	<0.001	−0.814	<0.001	−0.794	<0.001	−0.667	0.422
Cold exposure	0.916	<0.001	0.822	<0.001	0.822	<0.001	0.825	<0.001	0.812	0.304
Rainstorm exposure	1.444	<0.001	0.429	<0.001	0.429	0.008	0.435	<0.001	0.573	0.487
Air conditioners	0.319	0.008	0.853	<0.001	0.853	<0.001	0.841	<0.001	0.430	0.011
Heating systems	1.141	<0.001	0.606	<0.001	0.606	<0.001	0.607	<0.001	0.323	0.012
PCDI levels	1.176	<0.001	0.040	0.756	0.040	0.819	0.041	0.745	0.220	0.773

Note: LR, simple linear regression. MLR, multiple linear regression. LMM, multiple linear mixed regression. ^a^ refers to the MLR adjusted with HAC estimation. ^b^ and ^c^ refer to LMM with student alone and both student and city as random effects, respectively.

## Data Availability

The individual, de-identified participant data are available on reasonable request from the corresponding author.

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
