# Peer review of "Knowledge, Attitude, Risk Perception, and Health-Related Adaptive Behavior of Primary School Children towards Climate Change: A Cross-Sectional Study in China"

_ijerph, 2022, doi:10.3390/ijerph192315648_

Round 1

Reviewer 1 Report

This is an interesting study which examines knowledge, attitude, risk perception in relation to climate change and their interlinkages with health-related adaptive behavior in primary school children in China. I believe that it merits publication in IJERPH, but after some issues are addressed. More specifically:

1. Some terms should be clearly defined (conceptual and operational definitions) from the beginning of the manuscript, such as “health-related adaptive behavior”. Also maybe  “knowledge”, “attitude”, and “perceived risk” in relation to climate change. This is very important, inter alia for the development of the instrument to measure these parameters.

2. More information is needed in section 2.2. on the development and validation of the questionnaire: How many experts were used to review the questions? Was the instrument validated and how? Was there a pilot study conducted? Also, the studies used to develop each one of the 4 axes of the questionnaire could be further defined when referring to each axis (in the present form they are all listed together in the beginning of the section).

3. Figure S1 should move from the supplementary material to the main manuscript. Location of the meteorological stations may also be included in the figure (if these do not coincide with the cities).

4. The authors note (in the abstract and beyond) that (a) SEM showed that knowledge was positively associated with attitude and risk perception and (b) that knowledge, attitude, and risk perception corresponded to 31.6%, 22.8%, and 26.1% changes of adaptive behavior, respectively. How is the association between the three first variables affecting their predictive capacity of changes in the adaptive behaviour? The authors should elaborate on this (e.g. in section 2.4 or in the discussion of the results).

5. Section 3.1, ”The number of questionnaires ranged from 635 to 903.”: I presume the authors mean “per city”. This should be clarified in the text.

6. The conclusions section should be enhanced to showcase the added value of this manuscript, vis-à-vis other existing literature.

Reviewer 2 Report

Line 113: I assume that the term 'participants' refers to the parents of the pupils, since they are minors.

Please specify further.

Although the study was conducted on a significant sample and supported by a robust descriptive and correlation statistical analysis, as the Authors themselves indicated in the "Conclusions" as a weak point of the study, perplexity persists as to the reliability of the answers concerning the children's awareness: is the sample really aware of the significance of climate change and the short- and long-term consequences? Has the possibility that their answers are influenced by adult judgements been assessed? If so, in what way?

Reviewer 3 Report

The text is well written and the methodology is adequate.

Author Response

Response to Reviewer 3 Comments

Comment: The text is well written and the methodology is adequate.

Response: Thanks every much for your evaluation.

Round 2

Reviewer 1 Report

The authors adequately responded to this reviewer's comments